# A Plug-and-Play Image Registration Network

**Junhao Hu**,* **Weijie Gan**,* **Zhixin Sun, Hongyu An, Ulugbek S. Kamilov**
Washington University in St. Louis, St. Louis, MO, USA
{hjunhao,weijie.gan,zhixin.sun,hongyuan,kamilov}@wustl.edu

## Abstract

Deformable image registration (DIR) is an active research topic in biomedical imaging. There is a growing interest in developing DIR methods based on deep learning (DL). A traditional DL approach to DIR is based on training a convolutional neural network (CNN) to estimate the registration field between two input images. While conceptually simple, this approach comes with a limitation that it exclusively relies on a pre-trained CNN without explicitly enforcing fidelity between the registered image and the reference. We present *plug-and-play image registration network (PIRATE)* as a new DIR method that addresses this issue by integrating an explicit data-fidelity penalty and a CNN prior. PIRATE pre-trains a CNN denoiser on the registration field and *"plugs"* it into an iterative method as a regularizer. We additionally present PIRATE+ that fine-tunes the CNN prior in PIRATE using deep equilibrium models (DEQ). PIRATE+ interprets the fixed-point iteration of PIRATE as a network with effectively infinite layers and then trains the resulting network end-to-end, enabling it to learn more task-specific information and boosting its performance. Our numerical results on OASIS and CANDI datasets show that our methods achieve state-of-the-art performance on DIR.

## 1 Introduction

Deformable image registration (DIR) is an important component in modern medical imaging (Sotiras et al., 2013; Haskins et al., 2020; Maintz & Viergever, 1998). DIR seeks to estimate a dense registration field to align voxels between a moving image and a fixed image. DIR has become an active research area with many applications, including radiation therapy planning (Brock et al., 2010), disease progression tracking (Ashburner & Friston, 2000), and image-enhanced endoscopy (Uneri et al., 2013). DIR is often formulated as an optimization problem that minimizes an energy function composed of two terms: a penalty function measuring the similarity between the aligned image and the fixed image, and a regularizer imposing prior constraints on the registration field (e.g., smoothness via the gradient loss penalty). This optimization problem is usually solved using iterative methods (Thirion, 1998; Bajcsy & Kovačič, 1989; Rueckert et al., 1999; Glocker et al., 2008).

Deep learning (DL) has recently gained importance in DIR due to its promising performance (Haskins et al., 2020; Fu et al., 2020; Boveiri et al., 2020). A common approach in this context is to use a convolutional neural network (CNN) to directly estimate the registration fields between the moving and fixed images. The training process uses an energy function as the loss function. However, a potential limitation of this approach is that its results exclusively depend on the pre-trained CNN, without incorporating an explicit penalty function for imposing data consistency during inference. Several studies have explored the integration of explicit penalty and CNN priors in the inference phase by training a recursive network comprising multiple blocks that incorporate both explicit penalty and CNNs (Qiu et al., 2022; Jia et al., 2023; Wang et al., 2023b). In the context of inverse problems in imaging, such methods are often referred to as model-based deep learning (MBDL) (Ongie et al., 2020; Wen et al., 2023; Kamilov et al., 2023).

Plug-and-play (PnP) priors are widely used MBDL frameworks for solving inverse problems in imaging (Venkatakrishnan et al., 2013). The central idea of PnP is that one can use a pretrained neural network as a Gaussian denoiser as a regularizer for solving any inverse problem. The motivation behind using a denoiser prior is that training neural network denoisers can model the statistical

---

*Denotes co-first author.

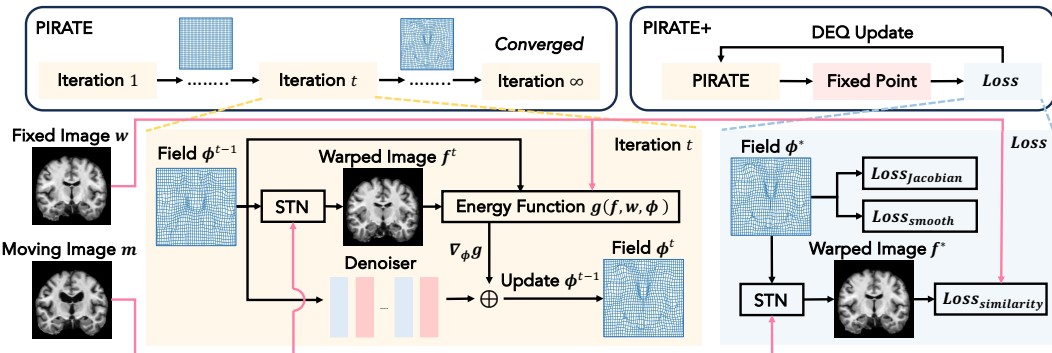

Figure 1: Illustration of the PIRATE and PIRATE+ pipelines. PIRATE updates the registration field $\phi$ using the penalty function that measures the similarity between the warped image and the fixed image, as well as a pre-trained CNN denoiser used as a regularizer. The DEQ update in PIRATE+ enables to fine-tune the CNN by calculating the gradients using implicitly differentiation through the fixed point of the forward iteration. As described in this paper, the DEQ update of PIRATE+ is computed using the weighted loss consisting of similarity loss, smoothness loss, and Jacobian loss.

distribution of high-dimensional data. PnP has been successfully used in many imaging applications such as super-resolution, phase retrieval, microscopy, and medical imaging (Nachaoui et al., 2021; Wu et al., 2019; Zhang et al., 2017b; Sun et al., 2019; Wei et al., 2020; Liu et al., 2020; Zhang et al., 2019) (see also reviews Ahmad et al. (2020); Kamilov et al. (2023)). Practical success of PnP has also motivated novel extensions, theoretical analyses, statistical interpretations, as well as connections to related approaches such as score matching and diffusion models (Chan et al., 2016; Romano et al., 2017; Buzzard et al., 2018; Reehorst & Schniter, 2018; Ryu et al., 2019; Kadkhodaie & Simoncelli, 2021; Cohen et al., 2021; Hurault et al., 2022a;b; Laumont et al., 2022; Gan et al., 2023a).

Despite the rich literature on PnP, the existing work on the topic has primarily focused on using denoisers to specify priors on images. To the best of our knowledge, the potential of denoisers to specify priors over registration fields in DIR has never been investigated before. We address this gap by proposing a new DIR method *Plug-and-play Image RegistrATion nEtwork (PIRATE)*. PIRATE is the first PnP approach that trains a CNN-based denoiser on the registration field and integrates this denoiser as a regularizer within iterative methods. We additionally present PIRATE+ as an extension of PIRATE based on deep equilibrium learning (DEQ) (Bai et al., 2019) that can fine-tune the denoiser to learn more task-specific information from the training dataset. PIRATE+ interprets the fixed-point iteration of PIRATE as a network with effectively infinite layers and trains the resulting network end-to-end by calculating implicit gradients based on fixed point convergence. We propose a three-term loss function in DEQ training: normalized cross-correlation (NCC), gradient loss, and Jacobian loss. We present an extensive validation of PIRATE and PIRATE+ on two widely used datasets: OASIS (Marcus et al., 2007) and CANDI (Kennedy et al., 2012). We used Dice similarity coefficient (DSC) and ratio of negative Jacobian determinant (JD) to evaluate the quality of registration. On both two datasets and metrics, the experimental results show that both PIRATE and PIRATE+ achieves state-of-the-art performance. This work thus addresses a gap in the current literature by providing a new application to PnP and introducing a new principled framework for infusing prior information on registration fields in DIR.

## 2 RELATED WORK

**Deformable image registration.** DIR refers to the process of obtaining a registration field $\phi$ that maps the coordinates of the moving image $m$ to those of the fixed image $f$ by comparing the content of the corresponding image (Sotiras et al., 2013; Haskins et al., 2020; Maintz & Viergever, 1998). DIR is typically formulated as an optimization problem

$$\hat{\phi} = \arg\min_{\phi} \{g(\boldsymbol{f}, \phi \circ \boldsymbol{m}) + r(\phi)\} \tag{1}$$

where $\phi \circ m$ represents aligned (warped) image obtained by using $\phi$ to warp $m$, $g$ is the penalty function measuring the dissimilarity between the aligned and fixed images, and $r$ is a regularizer imposing prior constraints on the registration field. Examples of $g$ in DIR include mean squared error (MSE), global cross-correlation (GCC), and normalized cross-correlation (NCC). A commonly used $r$ is the smoothness regularizer implemented by the gradient loss $\|\nabla\phi\|_2^2$ (Balakrishnan et al., 2019; Mok & Chung, 2020; Wu et al., 2022), where $\nabla\phi$ denotes to the gradient of $\phi$.

**Iterative algorithms.** Iterative algorithms are traditional approaches to solve the optimization problem in (1) on each image pair with satisfactory performance. Widely used examples include Demons (Thirion, 1998; Vercauteren et al., 2009), elastic type methods (Bajcsy & Kovačič, 1989; Davatzikos, 1997), b-splines based methods (Rueckert et al., 1999; Xie & Farin, 2004), and discrete methods (Glocker et al., 2008; 2011). The key idea of these methods is to update the registration field with the gradient of an energy function consisting of their unique design of the penalty function and regularizers. For example, Demons (Thirion, 1998) uses the first derivative of intensity difference and a Gaussian smoothing regularizer. Log-Demons (Vercauteren et al., 2009) improves Demons by using the second derivative. Its regularizer consists of a Gaussian fluid-like regularizer and a Gaussian diffusion-like regularizer.

**Deep learning.** DL in DIR has gained widespread attention over the last few years due to its excellent performance (Haskins et al., 2020; Fu et al., 2020; Boveiri et al., 2020). A traditional approach of DL is to train a CNN $n_{\theta}$ that models $\phi$ following $\phi = n_{\theta}(f, m)$, where $\theta$ are network parameters updated by training loss $e$ (Balakrishnan et al., 2019; Mok & Chung, 2020; De Vos et al., 2019; 2017; Eppenhof et al., 2018). This method usually involves a spatial transform network (STN) (Jaderberg et al., 2015) to ensure a differentiable warping operator $\circ$. For example, Voxelmorph (Balakrishnan et al., 2019) uses the U-net as $n_{\theta}$ to exploit features from the image pairs. The moving image is warped by using the output registration field in STN. Another approach, SYMNet (Mok & Chung, 2020), involves fully convolutional network (FCN) as $n_{\theta}$. It performs stronger topology preservation and invertibility by learning the symmetric deformation fields that also formulates the inverse transformation. Moreover, recent DL approaches such as reinforcement learning (Luo et al., 2022; Hu et al., 2021; Lotfi et al., 2013; Liu & Liu, 2022) and generative models (Kim et al., 2022; Tanner et al., 2018; Mahapatra et al., 2018; Zhang et al., 2020) have shown their potentials in DIR. However, the output of DL methods in the inference phase exclusively depends on the pre-trained CNN without incorporating the penalty function that can promote data consistency.

**Deep Unfolding.** DU is a DL paradigm with roots in sparse coding (Gregor & LeCun, 2010; Chen et al., 2018b) that has recently been used in DIR (Qiu et al., 2022; Jia et al., 2023; Wang et al., 2023b), due to its ability to provide a systematic connection between iterative algorithms and CNN architectures. For example, GraDIRN (Qiu et al., 2022) updates $\phi$ in iterations of form $\phi^{t+1} = \phi^t - \gamma(\nabla g(\phi^t) + \nabla n_{\theta}(f, m, \phi^t))$. A practical limitation of DU is that the number of iterations is usually limited due to the the high computational and memory complexity of end-to-end training. In GraDIRN, this limitation is reflected in its network structure that contains only 9 iterations.

**Plug-and-play.** PnP is a MBDL approach that has been extensively used for solving inverse problems in imaging (Venkatakrishnan et al., 2013; Romano et al., 2017; Kamilov et al., 2023). The key idea behind PnP is that one can use pre-trained deep denoisers to specify regularizers for various inverse problems. The key difference between PnP and DU is that PnP does not seek to train a MBDL architecture by instead focusing on an easier task of traininig a Gaussian. For example, a widely used variant of PnP is the steepest descent regularization by denoising (SD-RED) (Romano et al., 2017)

$$x^{t+1} = \mathsf{T}_{\theta}(x^t) = x^t - \gamma[\nabla g(x^t) + \tau(x^t - \mathsf{D}(x^t))] \tag{2}$$

where $g$ denotes the data-fidelity term that measures data consistency. The data-fidelity term is conceptually similar to the use of $g$ in (1) for DIR. The operator D denotes an additive white Gaussian noise (AWGN) denoiser (Zhang et al., 2017a; 2021; Rudin et al., 1992; Dabov et al., 2007) on $x$. It is worth noting that PnP based methods typically train the denoiser on images. Recent work has shown the potential of training the denoiser on other modalities. For example, recent work (Gan et al., 2023a) has explored the effectiveness of training the denoiser on the measurement operators, such as blur kernels. To the best of our knowledge, no prior work has explored using denosers within PnP to regularize registration fields in DIR.

**Deep equilibrium learning.** DEQ (Bai et al., 2019; Winston & Kolter, 2020; Gilton et al., 2021) and neural ordinary differential equations (NODE) (Chen et al., 2018a; Dupont et al., 2019; Kelly et al., 2020) are two related methods for training infinite-depth, weight-tied feedforward networks. Both of them have shown their effectiveness in inverse problems (Pramanik & Jacob, 2022; Pramanik et al., 2023; Liu et al., 2022; Zhao et al., 2022). The key idea of NODE is to represent the network with an infinite continuous-time perspective, while DEQ updates the network by analytically backpropagating through the fixed points using implicit differentiation. The DEQ output is specified implicitly as a fixed point of the operator $\mathsf{T}_{\boldsymbol{\theta}}$ parameterized by weights $\boldsymbol{\theta}$

$$\bar{\boldsymbol{x}} = \mathsf{T}_{\boldsymbol{\theta}}(\bar{\boldsymbol{x}}) \tag{3}$$

In forward pass, DEQ runs a fixed-point iteration to estimate $\bar{\boldsymbol{x}}$. The DEQ backward pass produces the gradients with respect to $\boldsymbol{\theta}$ by implicitly differentiating through the fixed point $\bar{\boldsymbol{x}}(\boldsymbol{\theta})$. For example, for the least-squares loss function, we get an analytical gradient

$$\ell(\boldsymbol{\theta}) = \frac{1}{2}\|\bar{\boldsymbol{x}}(\boldsymbol{\theta}) - \boldsymbol{x}^*\|_2^2 \quad \Rightarrow \quad \nabla\ell(\boldsymbol{\theta}) = (\nabla_{\boldsymbol{\theta}}\mathsf{T}_{\boldsymbol{\theta}}(\bar{\boldsymbol{x}}))^{\mathsf{T}}(\mathsf{I} - \nabla_{\boldsymbol{x}}\mathsf{T}_{\boldsymbol{\theta}}(\bar{\boldsymbol{x}}))^{-\mathsf{T}}(\bar{\boldsymbol{x}} - \boldsymbol{x}^*) \tag{4}$$

where $\boldsymbol{x}^*$ is the training label, and $\mathsf{I}$ is the identity mapping. Two recent papers have investigated DEQ and ODE in DIR. DEQ-RAFT (Bai et al., 2022) extends the recurrent flow estimator RAFT (Teed & Deng, 2020) into infinite layers and updates the resulting network using implicit DEQ differentiation. NODEO (Wu et al., 2022) models each voxel in moving images as a moving particle and considers the set of all voxels as a high-dimensional dynamical system whose trajectory determines the registration field. NODEO uses a neural network to represent the registration field and optimizes that dynamical system using adjoint sensitivity method (ASM) (Chen et al., 2018a) in NODE. The proposed PIRATE/PIRATE+ methods are different from these prior approaches since they combine an explicit penalty function $g$ and a CNN prior. DEQ-RAFT does not have any penalty function, while NODEO does not involve learning from training data.

**Our contribution.** *(1)* Our first contribution is the use of learned denoisers for regularizing the registration fields. The proposed PIRATE method can thus be seen as an extension of PnP framework to DIR. *(2)* Our second contribution is a new DEQ method for fine-tuning the regularizer within PnP iterations. The proposed PIRATE+ method thus extends prior work on DEQ for DIR by integrating both a penalty function measuring the dissimilarity between aligned and fixed images, and a CNN regularizer for infusing prior knowledge. *(3)* Our third contribution is the extensive validation of the proposed methods on OASIS and CANDI datasets. Our numerical results show the SOTA performance of PIRATE and PIRATE+, thus highlighting the potential of PnP and DEQ for DIR.

## 3 A PLUG-AND-PLAY IMAGE REGISTRATION NETWORK

In this section, we present two new methods: PIRATE and PIRATE+. Fig. 1 illustrates the pipelines of PIRATE and PIRATE+. PIRATE is a new PnP algorithm that integrates a pre-trained denoiser for estimating $\phi$. PIRATE+ fine-tunes the AWGN denoiser into a task-specific regularizer using DEQ. For each image pair during DEQ training, the forward iterations of PIRATE run until a fixed point. The backward DEQ iterations update the regularizer by backpropagating through the fixed point using implicit differentiation of the loss.

### 3.1 PIRATE

The objective function that PIRATE optimizes is formulated as a variation of (1)

$$\hat{\boldsymbol{\phi}} = \underset{\boldsymbol{\phi}}{\arg\min}\left\{g(\boldsymbol{f}, \boldsymbol{\phi} \circ \boldsymbol{m}) + h(\boldsymbol{\phi})\right\} \tag{5}$$

where $h$ represents the regularizer consisting of an explicit smoothness constraint and the implicit denoising regularizer. The iterations of PIRATE can be expressed as

$$\boldsymbol{\phi}^{t+1} \leftarrow \boldsymbol{\phi}^t - \gamma\left(\nabla g(\boldsymbol{f}, \boldsymbol{\phi}^t \circ \boldsymbol{m}) + \alpha\nabla r(\boldsymbol{\phi}^t) + \tau(\boldsymbol{\phi}^t - \mathsf{D}(\boldsymbol{\phi}^t))\right) \tag{6}$$

where $\boldsymbol{\phi}^t$ is the estimated registration field at iteration $t$, $\gamma > 0$ is a step size, and $\alpha > 0$ and $\tau > 0$ are regularization parameters.

The PIRATE updates consist of three terms. The first term $g$ is the penalty function measuring similarity between the aligned and fixed images, which in our implementation corresponds to the global cross correlation (GCC) function

$$g(\boldsymbol{f}, \boldsymbol{\phi} \circ \boldsymbol{m}) = 1 - \frac{1}{\sigma_{\boldsymbol{f}}\sigma_{\boldsymbol{\phi}\circ\boldsymbol{m}}} \sum (\boldsymbol{f} - \mu_{\boldsymbol{f}}) \odot (\boldsymbol{\phi} \circ \boldsymbol{m} - \mu_{\boldsymbol{\phi}\circ\boldsymbol{m}}), \qquad (7)$$

where $\mu$ and $\sigma$ denote the mean and the standard deviation of the warped image $\boldsymbol{\phi} \circ \boldsymbol{m}$ and the fixed image $\boldsymbol{f}$. The second term is the classical smoothness promoting regularizer $r(\boldsymbol{\phi}) := \|\nabla \boldsymbol{\phi}\|_2^2$.

The third term is the PnP regularizer consisting of the residual term $\boldsymbol{x} - \mathsf{D}(\boldsymbol{x})$, where $\mathsf{D}$ is a pre-trained AWGN denoiser. We will consider denoisers trained to perform minimum mean squared error (MMSE) estimation of $\boldsymbol{\phi}$ from

$$\boldsymbol{z} = \boldsymbol{\phi} + \boldsymbol{n} \quad \text{with} \quad \boldsymbol{\phi} \sim p_{\boldsymbol{\phi}}, \quad \boldsymbol{n} \sim \mathcal{N}(\boldsymbol{0}, \sigma^2 \mathbf{I}). \qquad (8)$$

MMSE denoisers can be analytically expressed as

$$\mathsf{D}(\boldsymbol{z}) = \mathbb{E}[\boldsymbol{\phi}|\boldsymbol{z}] = \int \boldsymbol{\phi}\, p_{\boldsymbol{\phi}|\boldsymbol{z}}(\boldsymbol{\phi}; \boldsymbol{z})\, \mathsf{d}\boldsymbol{\phi} = \int \boldsymbol{\phi}\, \frac{G_{\sigma}(\boldsymbol{z} - \boldsymbol{\phi}) p_{\boldsymbol{\phi}}(\boldsymbol{\phi})}{p_{\boldsymbol{z}}(\boldsymbol{z})}\, \mathsf{d}\boldsymbol{\phi}, \qquad (9)$$

where we used the probability density function of the noisy registration field

$$p_{\boldsymbol{z}}(\boldsymbol{z}) = \int G_{\sigma}(\boldsymbol{z} - \boldsymbol{\phi}) p_{\boldsymbol{\phi}}(\boldsymbol{\phi})\, \mathsf{d}\boldsymbol{\phi}. \qquad (10)$$

The function $G_{\sigma}$ in (9) denotes the Gaussian density with the standard deviation $\sigma > 0$. The remarkable property of the MMSE denoiser is that its residual satisfies (Robbins & Monro, 1951)

$$\boldsymbol{\phi} - \mathsf{D}(\boldsymbol{\phi}) = -\sigma^2 \log p_{\boldsymbol{z}}(\boldsymbol{\phi}), \qquad (11)$$

which implies that PIRATE is seeking to compute high probability registration fields from $p_{\boldsymbol{z}}$, which is a Gaussian smoothed version of the prior $p_{\boldsymbol{\phi}}$.

## 3.2 PIRATE+

PIRATE+ uses DEQ to fine-tune the regularizer $\mathsf{D}$ in the PIRATE iteration by minimizing the following loss

$$\ell(\boldsymbol{\phi}, \boldsymbol{f}, \boldsymbol{m}) = w_0 \ell_{\mathsf{smi}}(\boldsymbol{f}, \boldsymbol{\phi} \circ \boldsymbol{m}) + w_1 \ell_{\mathsf{smt}}(\boldsymbol{\phi}) + w_2 \ell_{\mathsf{Jac}}(\boldsymbol{\phi}) \qquad (12)$$

where $w_0$, $w_1$, and $w_2$ are non-negative weights. The term $\ell_{\mathsf{smt}}$ in (12) corresponds to the same smoothness regularizer in (6). The term $\ell_{\mathsf{smi}}$ is the normalized cross correlation (NCC) function

$$\ell_{\mathsf{smi}}(\boldsymbol{f}, \boldsymbol{\phi} \circ \boldsymbol{m}) = 1 - \frac{1}{n}\sum_{\boldsymbol{r}} \frac{1}{\sigma_{\boldsymbol{f}(\boldsymbol{r})}\sigma_{\boldsymbol{\phi}\circ\boldsymbol{m}(\boldsymbol{r})}} \sum (\boldsymbol{f}(\boldsymbol{r}) - \mu_{\boldsymbol{f}(\boldsymbol{r})}) \odot (\boldsymbol{\phi} \circ \boldsymbol{m}(\boldsymbol{r}) - \mu_{\boldsymbol{\phi}\circ\boldsymbol{m}(\boldsymbol{r})}), \quad (13)$$

where $\boldsymbol{f}(\boldsymbol{r})$ and $\boldsymbol{\phi} \circ \boldsymbol{m}(\boldsymbol{r})$ refer to the 3D sliding windows centered on the pixel $\boldsymbol{r} = (x, y, z)$ in the fixed image and the aligned image, respectively. The quantities $\mu$ and $\sigma$ denote the mean and the standard deviation in those windows. The term $\ell_{\mathsf{Jac}}$ in (12) represents Jacobian loss

$$\ell_{\mathsf{Jac}}(\boldsymbol{\phi}) = \frac{1}{n}\sum_{i} \|\mathsf{ReLU}(-|J(i)|)\|_2^2. \qquad (14)$$

where $|J(i)|$ denotes the *Jacobian determinant (JD)* at each $i$ in $\boldsymbol{\phi}$. Negative JD in registration field indicates non-physical transformations such as flipping. A value greater than one indicates local expansion, a value less than one but greater than zero indicates local compression, and a negative value indicates a physically implausible inversion. The $\ell^2$-norm promotes the sparsity on negative JD by filtering positive values using ReLU activation function, leading to physically plausible transformations in DIR. Note that regularizer in PIRATE+ is learned by minimizing *the loss of a specific registration task*. On the other hand, regularizer in PIRATE is learned by minimizing a MSE loss of purely denoising noisy registration fields.

We adopt Jacobian-Free DEQ (JFB) (Fung et al., 2022) to efficiently minimize the PIRATE+ loss function

$$\nabla \ell_{\mathsf{JFB}}(\boldsymbol{\theta}) = (\nabla_{\boldsymbol{\theta}} \mathsf{T}_{\boldsymbol{\theta}}(\bar{\boldsymbol{\phi}}))^{\mathsf{T}} (\nabla_{\bar{\boldsymbol{\phi}}} \ell(\bar{\boldsymbol{\phi}}(\boldsymbol{\theta}), \boldsymbol{f}, \boldsymbol{m})) \qquad (15)$$

where $\nabla \ell_{\mathsf{JFB}}(\boldsymbol{\theta})$ is been a descent direction for the loss function $\ell$ with respect to $\boldsymbol{\theta}$ (Fung et al., 2022). The effectiveness of JFB has been shows in prior work (Wang et al., 2023a; Gan et al., 2023b; Li et al., 2022).

Table 1: Numerical results of DSC, ratio of negative JD, and inference time for PIRATE, PIRATE+, and benchmarks on OASIS and CANDI datasets. The variances are shown in parentheses. Note that the **best** and the second-best result are highlighted in red and blue. The result shows that PIRATE+ performs state-of-the-art performance comparing with other baselines. Moreover, PIRATE can achieve competitive performance without DEQ

| Method | OASIS | | CANDI | | Time (s) |
|---|---|---|---|---|---|
| | Avg. DSC ↑ | Neg. JD % ↓ | Avg. DSC ↑ | Neg. JD % ↓ | |
| SyN | 0.6986 (0.028) | 0.1471 (0.002) | 0.7274 (0.049) | 0.1211 (0.001) | 42.52 |
| VoxelMorph | 0.7655 (0.033) | 0.0940 (0.001) | 0.7459 (0.014) | 0.1149 (0.020) | **0.17** |
| SYMNet | 0.7664 (0.032) | 0.0796 (0.001) | 0.7489 (0.016) | 0.0912 (0.001) | 0.32 |
| GraDIRN | 0.7311 (0.054) | 0.1418 (0.002) | 0.7362 (0.019) | 0.1532 (0.028) | 0.49 |
| Log-Demons | 0.7931 (0.040) | 0.0912 (0.007) | 0.7553 (0.036) | 0.0997 (0.006) | 46.72 |
| NODEO | 0.7947 (0.027) | 0.0531 (0.001) | 0.7629 (0.019) | 0.0726 (0.001) | 89.01 |
| PIRATE | 0.7948 (0.033) | 0.0567 (0.002) | 0.7624 (0.020) | 0.0833 (0.002) | 32.01 |
| PIRATE+ | **0.7952 (0.029)** | **0.0495 (0.001)** | **0.7633 (0.027)** | **0.0660 (0.001)** | 49.47 |

## 4 NUMERICAL EXPERIMENTS

### 4.1 SETUP

**Datasets.** We validated PIRATE and PIRATE+ on two widely used datasets: OASIS-1 (Marcus et al., 2007) and CANDI (Kennedy et al., 2012). The OASIS-1 dataset includes a cross-sectional collection of 416 subjects with T1-weighted brain MRI scans and anatomical segmentations. We followed a widely used preprocessing method[1], including affine pre-registration, skull stripping, and rescaling to 0-1. All images were padded to the size [192, 128, 224]. We used anatomical segmentations of 35 different brain structures provided by OASIS-1 for evaluation. The CANDI dataset comprises 54 T1-weighted brain MRI scans and anatomical segmentations. The preprocessing does not include affine pre-registration and skull stripping, since images in CANDI are pre-registered and do not contain skulls. Besides the preprocessing methods used on OASIS, we excluded structures smaller than 1000 pixels in anatomical segmentations and used the remaining 28 structures for evaluation in CANDI. For both datasets, we randomly shuffled the images and allocated 100 unique image pairs for training and another 100 unique image pairs for evaluation.

**Evaluation metrics.** We adopted two evaluation metrics: Dice similarity coefficient (DSC) and ratio of negative JD, to evaluate the accuracy and quality of the registration. The DSC quantifies the spatial overlap between the anatomical segmentations of the fixed image and the warped image. We obtained the segmentation mask of the warped image by applying the associated registration field to the segmentation mask of the moving image. JD evaluates physical plausibility of the transformation by quantifying the expansion and compression of each voxel's neighbors. We employed the ratio of negative JD of each registration field to evaluate the quality of the transformation.

### 4.2 IMPLEMENTATION DETAILS

**PIRATE setup.** The penalty function in PIRATE is based on GCC, which, unlike NCC, relies on the whole image instead of sliding windows. Due to lower computational complexity of GCC, we used it instead of NCC in the forward iterations. In our implementation, we used a dynamic cosine-based step size in (6)

$$\gamma^t = \frac{1}{2}\gamma^0(1 + \cos(\frac{\pi t}{t_{\max}}))  \tag{16}$$

which we observed to give better results compared to the fixed $\gamma$.

We chose DnCNN (Rudin et al., 1992) as a deep learning based AWGN denoiser. We obtained the ground truth for DnCNN training by using registration fields estimated by NODEO. In training

---

[1]https://github.com/adalca/medical-datasets

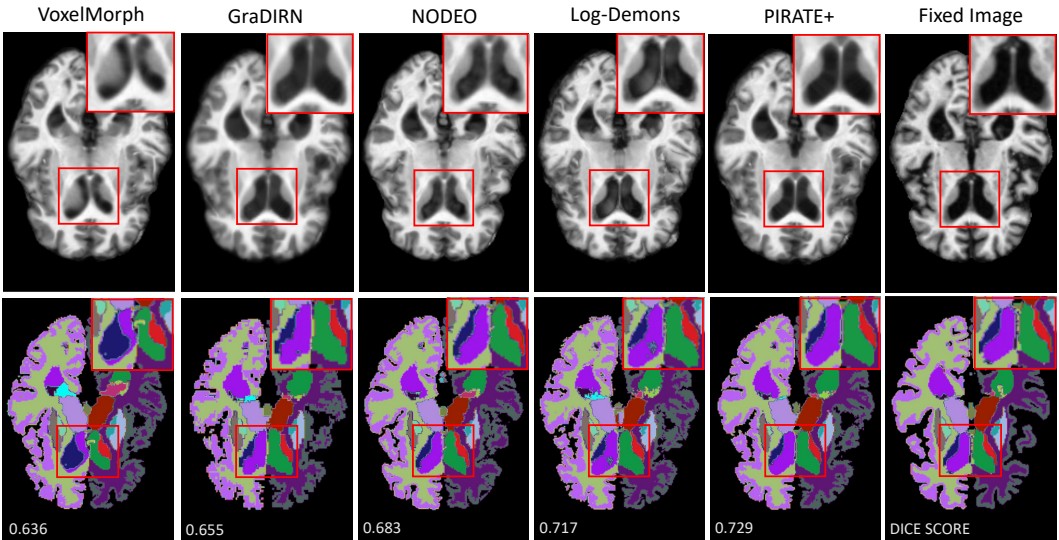

Figure 2: The visual results of warped images (top) and correlated warped segmentation masks (bottom) from PIRATE+ and selected benchmarks on the OASIS dataset. The regions of interest are highlighted within a red box. The DSC for each image is displayed in the bottom row. Note that the result of PIRATE+ is more consistent with the fixed image with fewer artifacts compared with other baselines.

Table 2: Numerical results of DSC, Jacobian determinant, and inference time for PIRATE+ and its 5 ablated variants. The variances are shown in parentheses. We denote P as the penalty function, R as the denoiser regularizer, S as the smoothness regularizer, and D as the DEQ refined denoiser. In this case, PIRATE is formulated as P+R+S, and PIRATE+ is formulated as P+D+S. Note that the **best** and the second-best result are highlighted in red and blue. The result shows that the dual-regularizer structure and DEQ significantly reduce the negative Jacobian.

| Method | OASIS | | CANDI | | Time (s) |
|---|---|---|---|---|---|
| | Avg. DSC ↑ | JD ↓ | Avg. DSC ↑ | JD ↓ | |
| P | 0.7281 (0.044) | 6.2931 (0.002) | 0.7156 (0.047) | 7.5804 (0.003) | **20.20** |
| P+R | 0.7989 (0.023) | 1.3319 (0.002) | 0.7644 (0.001) | 2.1713 (0.031) | 28.37 |
| P+S | 0.7782 (0.012) | 0.2091 (0.001) | 0.7492 (0.019) | 1.5847 (0.001) | 25.64 |
| P+D | 0.7897 (0.008) | 0.1120 (0.001) | 0.7585 (0.011) | 1.1844 (0.001) | 27.64 |
| PIRATE: P+R+S | 0.7948 (0.033) | 0.0567 (0.002) | 0.7624 (0.020) | 0.0833 (0.002) | 32.01 |
| PIRATE+: P+D+S | 0.7952 (0.029) | 0.0495 (0.001) | 0.7633 (0.027) | 0.0660 (0.001) | 49.47 |

phase of the denoiser, we used Adam (Kingma & Ba, 2014) optimizer with learning rate $1e^{-4}$ for 400 epochs. We selected the best-performing denoiser from 10 denoisers trained on different noise levels (Gaussian noise with standard deviations starting from 1 to 10). Moreover, we downsampled the registration field to half the size of the image. The downsampled registration fields were interpolated back to the original size before being applied to the moving image. Note that the denoiser was also trained on the downsampled data. We tested different values of $\gamma^0$, $\alpha$, and $\tau$ in (6). PIRATE achieved the best performance by assigning $\gamma^0$ to $5 \times 10^5$, $\alpha$ to $5 \times 10^{-1}$, and $\tau$ to $1 \times 10^{-7}$ for OASIS-1. $\alpha$ was assigned to $5 \times 10^{-1}$ for CANDI while $\gamma^0$ and $\alpha$ stayed the same.

**DEQ training in PIRATE+.** We used the Anderson solver (Anderson, 1965) to accelerate the fixed-point iteration in PIRATE. For DEQ in PIRATE+, we used Adam optimizer with learning rate $1 \times 10^{-5}$ for 50 epochs. We assigned $w_0$ to 1, $w_1$ to 5, $w_2$ to 1 for both datasets. Note that only deep learning based denoiser (DnCNN in our experiment) can be fine-tuned by DEQ.

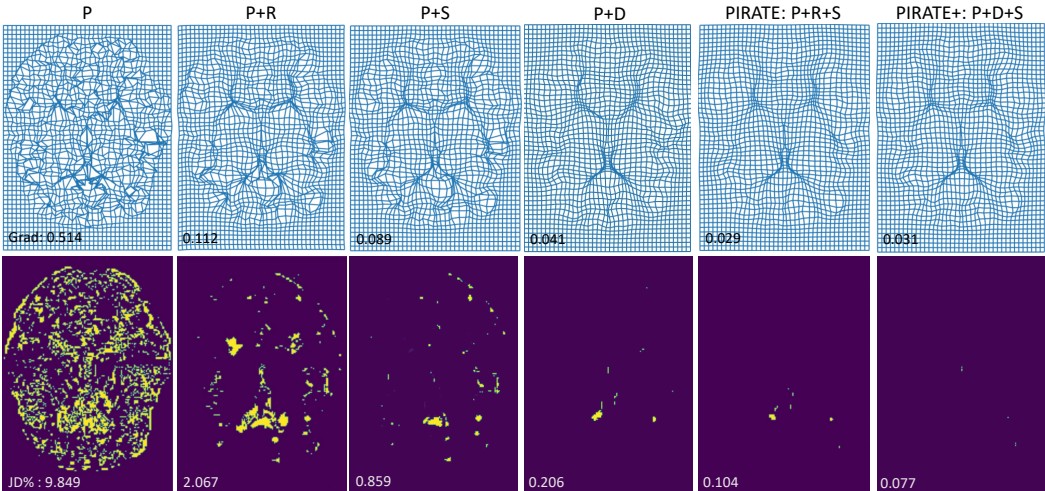

Figure 3: The visual results for the warped grid (top) and negative JD (yellow points in bottom) of PIRATE+ and its five ablated variants on CANDI dataset. We denote P as the penalty function, R as the denoiser regularizer, S as the smoothness regularizer, and D as DEQ. PIRATE is formulated as P+R+S, and PIRATE+ is formulated as P+D+S. The gradient loss is shown in the top row, and the ratio of negative JD is shown in the bottom row. Note that PIRATE's architecture is optimal, since it significantly reduces the negative JD and provides smoother registration field.

Table 3: Numerical results of DSC, Jacobian determinant, and inference time for PIRATE+ with fixed and dynamic step size in its forward iteration on OASIS and CANDI datasets. The variances are shown in parentheses. Note that the **optimal** results are highlighted in red. The result shows that the dynamic step size in the forward iteration improves the overall performance.

| Method | OASIS | | CANDI | | Time (s) |
|---|---|---|---|---|---|
| | Avg. DSC ↑ | JD ↓ | Avg. DSC ↑ | JD ↓ | |
| Fixed $\gamma$ | 0.7902 (0.044) | 0.0742 (0.001) | 0.7556 (0.071) | 0.0988 (0.002) | **31.42** |
| Dynamic $\gamma$ | 0.7952 (0.029) | 0.0495 (0.001) | 0.7633 (0.027) | 0.0660 (0.001) | 32.01 |

## 4.3 RESULTS

**Comparison with benchmarks.** We compared PIRATE and PIRATE+ against three iterative methods (SyN (Avants et al., 2008), Log-Demons (Vercauteren et al., 2009), and NODEO (Wu et al., 2022)), two CNN based methods (VoxelMorph (Balakrishnan et al., 2019) and SYMNet (Mok & Chung, 2020)), and one DU based method (GraDIRN (Qiu et al., 2022)). We retrained all DL based methods and DU based method using the optimal hyperparameters mentioned in their papers.

Table 1 shows the numerical results of PIRATE, PIRATE+ and benchmarks on OASIS and CANDI datasets. On both two datasets and metrics, PIRATE+ shows superior performance compared with other benchmarks. It is worth noting that PIRATE+ outperforms NODEO, although we used NODEO to generate the ground truth for the denoiser. The results of PIRATE show that the pre-trained AWGN denoiser can provide competitive results without refinement of the DEQ. Moreover, PIRATE+ performs competitive inference time while maintaining state-of-the-art performance compared with iterative baselines.

Fig. 2 shows the visual results of warped images (top) and correlated warped segmentation masks (bottom) from PIRATE+ and benchmarks on the OASIS dataset. In the top row, the result of PIRATE+ in the region of interest (ROI) is more consistent with the fixed image with fewer artifacts than other benchmarks. In the bottom row, the DSC shows that PIRATE+ achieves a better structure

Figure 4: The visualization and plot for the warped image from PIRATE+ with different iterations. The error map of ROI is shown at the upper right corner. The results show a convergence behaviour of PIRATE+.

matching. The ROI in the output of PIRATE+ shows better alignments to the fixed image while maintaining the original anatomical structure.

**Ablation study.** We conducted an ablation study to demonstrate the effectiveness of each part in PIRATE+. We denoted P as the penalty function, R as the denoiser regularizer, S as the smoothness regularizer, and D as DEQ refined denoiser regularizer. Note that PIRATE and PIRATE+ can be expressed as $P + S + R$ and $P + S + D$, respectively.

Table 2 shows the numerical results of PIRATE+ and its five ablated variants. The results show that the dual-regularizer structure and DEQ of PIRATE+ significantly reduce the ratio of negative JD with acceptable decrement in DSC.

Fig. 3 visualizes the warped grid (top) and negative JD (yellow points in bottom) from PIRATE+ and its five different variants. It is evident that the two regularizers and DEQ contribute to smooth registration fields while maintaining low ratio of negative JD, which underlines the effectiveness of the PIRATE+'s architecture.

**Convergence.** We designed an experiment to validate the convergence of PIRATE+. Fig. 4 illustrates the warped images from PIRATE+ with different iterations on CANDI dataset. The result shows a convergence behavior.

**Step size.** We tested the effectiveness of the dynamic step size compared with the fixed step size. Table 3 shows the numerical results of using fixed step size and dynamic step size in PIRATE+ on OASIS and CANDI datasets. The result shows that the dynamic step size improves the overall performance of PIRATE+.

## 5 CONCLUSION

This paper presents PIRATE as the first PnP-based method for DIR and PIRATE+ as its further DEQ extension. The key advantages of PIRATE and PIRATE+ are as follows: *(a)* PIRATE achieves competitive performance in comparison to other baselines. This superior performance is attributed to its ability to seamlessly integrate the penalty function with learned CNN priors; *(b)* PIRATE treats the CNN prior as an implicit regularizer, which renders it compatible with other regularizers. For instance, in our experiments, we combined it with the gradient loss to further enhance its performance; *(c)* PIRATE+ achieves state-of-the-art performance in comparison to other baselines. DEQ in PIRATE+ fine-tuned the denoiser in PIRATE, enabling it to learn more task-specific information. The training time of PIRATE+ is longer than traditional DL methods. However, our results show that PIRATE+ can gain noticeable improvement than traditional CNN-based methods. It's also worth mentioning that our validation of PIRATE and PIRATE+ in this work was based on brain MRI datasets. Potential future endeavors might explore their adaptability to different anatomies and imaging modalities, such as whole-body registration and cross-modal registration.

## 6 ACKNOWLEDGEMENT

This work was supported by the following grants: NIH R01HL129241, RF1NS116565, R21NS127425, R01 EB032713, and the NSF CAREER award under grants CCF-2043134.

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

# A    APPENDIX

Our main manuscript presents PIRATE and PIRATE+ as the first PnP based method in DIR. Our experimental results show the effectiveness of the architectures of PIRATE and PIRATE+ and demonstrate their state-of-the-art performance. The additional materials in this section supplement our experimental results and further support the conclusions mentioned above. In appendix, we provide the visualization of warped images and segmentation masks on CANDI dataset as a supplement to the results on OASIS dataset in main manuscript. We also provide additional results on the EMPIRE10 lung CT dataset. In addition to the image similarity and DSC used in main manuscript, we illustrate the warped grids on OASIS and CANDI datasets to further evaluate the performance using the smoothness of registration fields. We design an experiment by using total variation (TV) denoiser (Rudin et al., 1992) in PIRATE to demonstrate the compatibility of PIRATE with non-learning based denoisers. Moreover, we show the visualization of warped images and segmentation masks from PIRATE using dynamic and fixed step size on OASIS dataset, which matches the numerical results in main manuscript. To further evaluate our methods, we include the analysis of computational efficiency and scalability. To clarify the acronyms, we provide a table of acronyms and corresponding full names in this paper. We also provide the corner case of our method and the number of function evaluation(NFE) along with the training iterations of PIRATE+.

Fig. 5 shows the visual results for warped images and correlated warped segmentation masks from PIRATE+ and benchmarks on the CANDI dataset. In the top and bottom row, the results of PIRATE+ in ROI are more consistent with the fixed image with fewer artifacts than other benchmarks.

Table. 4 shows the numerical results of PIRATE, PIRATE+ and benchmarks on the EMPIRE10 lung CT dataset. PIRATE+ shows superior performance compared with other benchmarks. The results of PIRATE show that the pre-trained AWGN denoiser can provide competitive results without refinement of the DEQ.

Fig. 6 and Fig. 7 show the illustration of warped images and correlated warped segmentation masks from PIRATE+ and benchmarks on the EMPIRE10 lung CT dataset. In both warped images and segmentation masks, the results of PIRATE+ are more consistent with the reference than other benchmarks.

Fig. 8 shows the visual results of warped grid from OASIS and CANDI datasets of PIRATE and different baselines. The results show that PIRATE maintains the smoothness of the registration fields.

Fig. 9 shows the visual results of using dynamic step size and fixed step size from OASIS and CANDI datasets. It shows that dynamic step size can provide results more similar to the fixed images and segmentation masks with less artifact.

Fig. 10 shows the visual results of using TV denoiser and DnCNN denoiser from OASIS dataset. Table 5 shows the numerical result of using TV denoiser and DnCNN denoiser from OASIS and CANDI datasets. The results show that TV denoiser can still provide reasonable result, although with some performance degradation.

Table. 6 shows the numerical results of memory cost, running time in training and inference for PIRATE, PIRATE+, and benchmarks on OASIS and CANDI datasets. Table. 6 shows that PIRATE and PIRATE+ has lower memory complexity compared to other DL baselines.

Fig. 11 shows the visual results of memory cost and inference time of PIRATE and PIRATE+ with different size of inputs. Fig. 11 shows that PIRATE and PIRATE+ have similar scalability.

Table. 7 shows acronyms and corresponding full names in this paper.

Fig. 12 shows the visual results of the corner case of PIRATE+ (with the least improvement in Dice) compared with results from other benchmarks on this case. Note that the result of PIRATE+ still maintains state-of-the-art performance in the corner case.

Fig. 13 shows the visual results of NFE along with the training iterations of PIRATE+. The figure shows that the NFE converges in the training process.

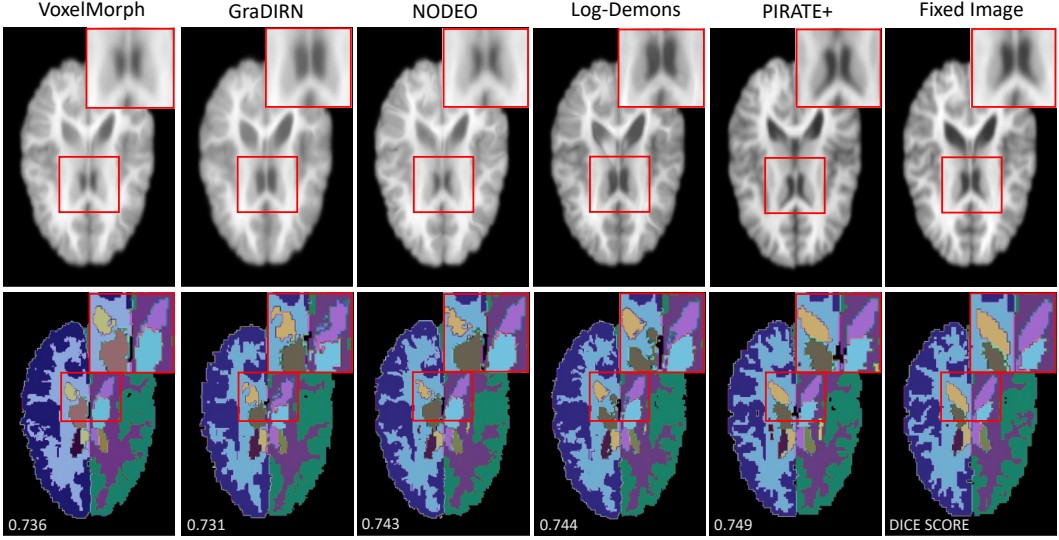

Figure 5: The visual results of warped images (top) and correlated warped segmentation masks (bottom) from PIRATE+ and selected benchmarks on the CANDI dataset. The regions of interest are highlighted within a red box. The DSC for each image is displayed in the bottom row. Note that the result of PIRATE+ is more consistent with the fixed image with fewer artifacts comparing with other baselines.

Table 4: Numerical results of DSC, ratio of negative JD, and inference time for PIRATE, PIRATE+, and benchmarks on the EMPIRE10 lung CT dataset. The variances are shown in parentheses. Note that the **best** and the second-best result are highlighted in red and blue. The result shows that PI-RATE+ performs state-of-the-art performance comparing with other baselines. Moreover, PIRATE can achieve competitive performance without DEQ

| Method | Avg. DSC ↑ | Neg. JD % ↓ | Time (s) |
|---|---|---|---|
| SyN | 0.9246 (0.045) | 0.0854 (0.008) | 37.28 |
| VoxelMorph | 0.9681 (0.022) | 0.0521 (0.012) | **0.16** |
| SYMNet | 0.9701 (0.012) | 0.0542 (0.009) | 0.28 |
| GraDIRN | 0.9644 (0.012) | 0.0542 (0.009) | 0.44 |
| Log-Demons | 0.9772 (0.019) | 0.0571 (0.011) | 43.29 |
| NODEO | 0.9802 (0.028) | **0.0441 (0.008)** | 82.17 |
| PIRATE | 0.9797 (0.032) | 0.0546 (0.009) | 29.77 |
| PIRATE+ | **0.9811 (0.027)** | 0.0467 (0.007) | 45.32 |

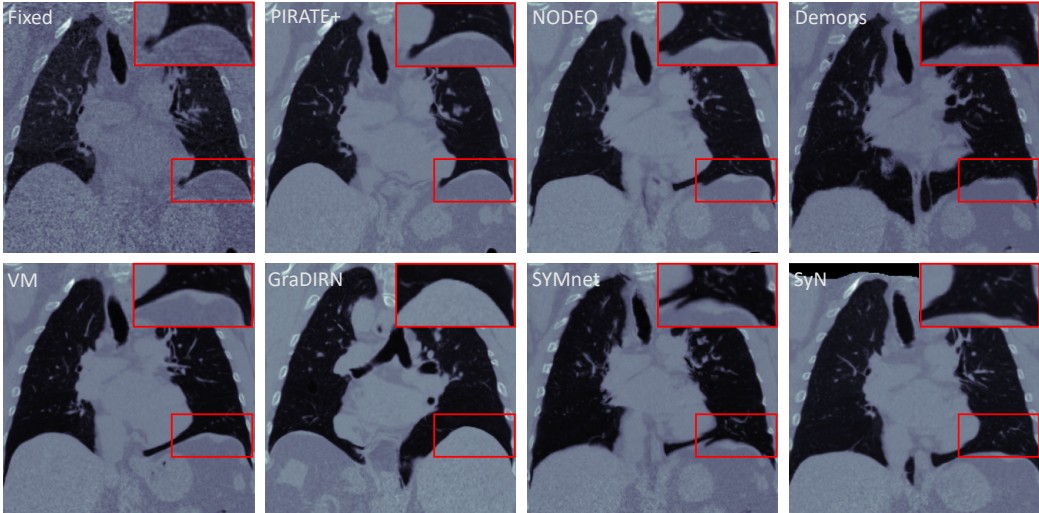

Figure 6: The visual results of warped images from PIRATE+ and selected benchmarks on the EMPIRE10 lung CT dataset. The regions of interest are highlighted within a red box. Note that the result of PIRATE+ is more consistent with the fixed image with fewer artifacts comparing with other baselines.

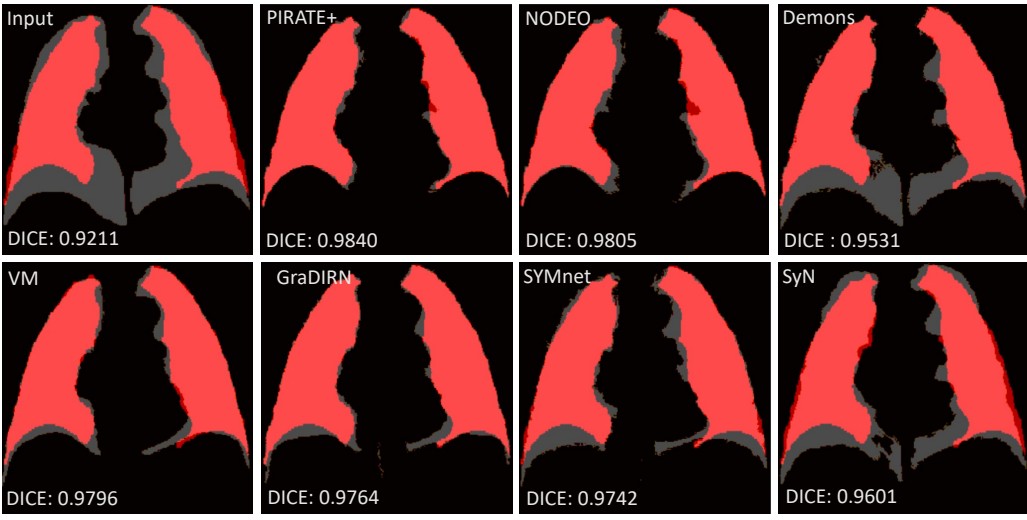

Figure 7: The visual results of warped segmentation masks from PIRATE+ and selected benchmarks on the EMPIRE10 lung CT dataset. The red region is the mask from fixed image, and the gray region is the misalignment of the warped mask. The DSC for each image is displayed in the bottom. Note that the result of PIRATE+ is more consistent with the fixed mask comparing with other baselines.

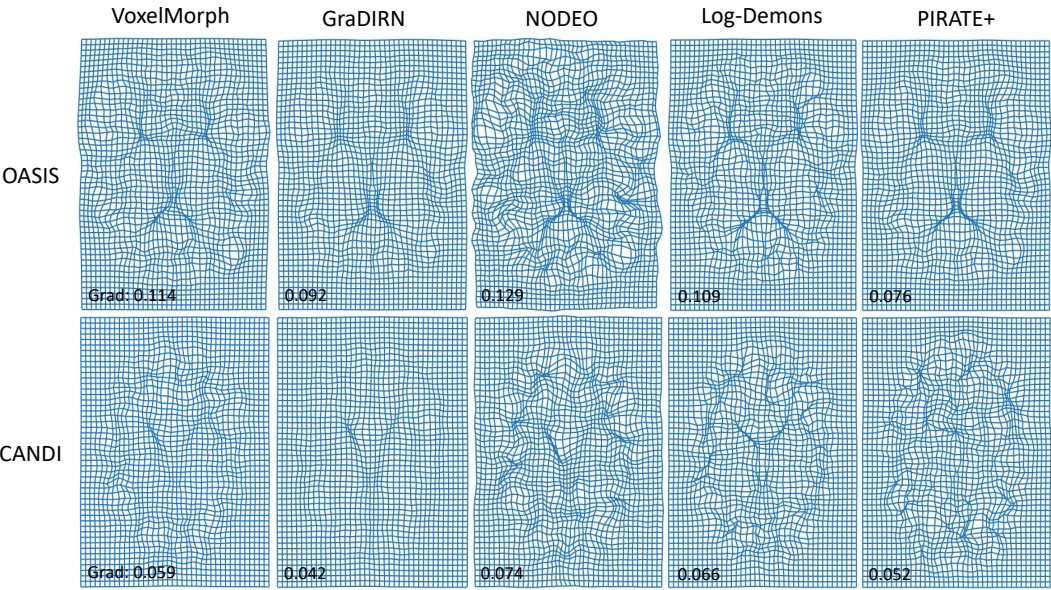

Figure 8: The visual results of warped grid from OASIS (top) and CANDI (bottom) of PIRATE+ and different baselines. The gradient loss is shown at the bottom of each image.

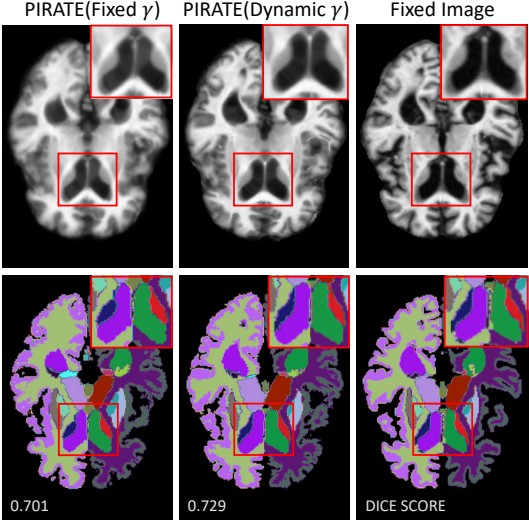

Figure 9: The visual results of warped images (top) and correlated warped segmentation masks (bottom) from PIRATE(fixed $\gamma$) and PIRATE(dynamic $\gamma$) on the OASIS dataset. The regions of interest are highlighted within a red box. The DSC for each image is displayed in the bottom row.

Table 5: Numerical results of DSC, Jacobian determinant, and inference time for PIRATE with total variation (TV) denoiser and DnCNN denoiser on OASIS and CANDI datasets. The variances are shown in parentheses. Note that the **optimal** results are highlighted in red.

| Method | OASIS | | CANDI | | Time (s) |
|---|---|---|---|---|---|
| | Avg. DSC ↑ | JD ↓ | Avg. DSC ↑ | JD ↓ | |
| PIRATE(TV) | 0.7617 (0.054) | 0.1281 (0.002) | 0.7481 (0.049) | 0.0992 (0.001) | 118.20 |
| PIRATE(DnCNN) | 0.7948 (0.033) | 0.0567 (0.002) | 0.7624 (0.020) | 0.0833 (0.002) | **32.01** |

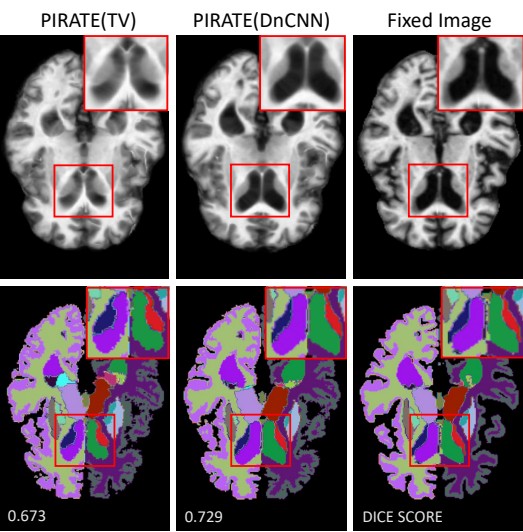

Figure 10: The visual results of warped images (top) and correlated warped segmentation masks (bottom) from PIRATE(TV) and PIRATE(DnCNN) on the OASIS dataset. The regions of interest are highlighted within a red box. The DSC for each image is displayed in the bottom row.

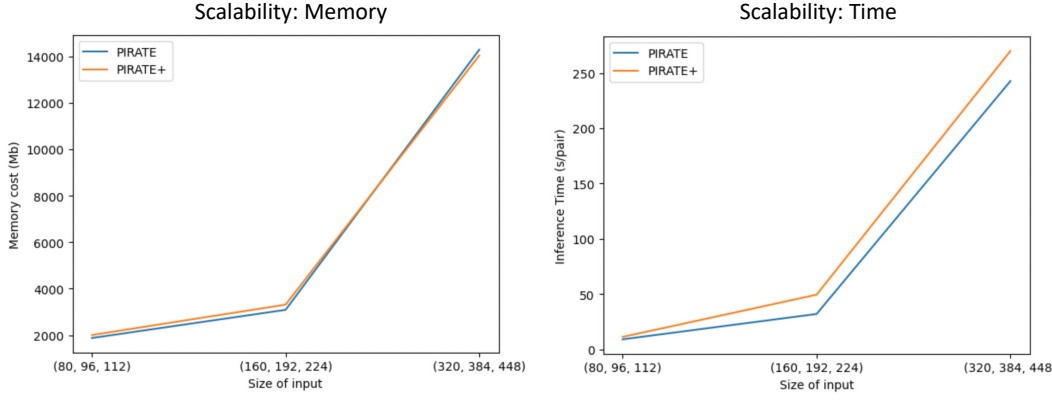

Figure 11: The illustrations of memory cost and inference time of PIRATE and PIRATE+ with different size of inputs. This figure shows that PIRATE and PIRATE+ have similar scalability.

Table 6: Numerical results of memory cost, running time in training and inference for PIRATE, PIRATE+, and benchmarks on OASIS and CANDI datasets. This table shows that PIRATE and PIRATE+ has lower memory complexity compared to other DL baselines.

| Method | Inference | | Training | |
|---|---|---|---|---|
| | Memory (MB) ↓ | Running time (s) ↓ | Memory (MB) ↓ | Running time (s/epoch) ↓ |
| SyN | 2252 | 42.52 | \ | \ |
| VoxelMorph | 7642 | 0.17 | 12846 | 78 |
| SYMNet | 5976 | 0.32 | 16448 | 185 |
| GraDIRN | 4386 | 0.49 | 11642 | 112 |
| Log-Demons | 2709 | 46.72 | \ | \ |
| NODEO | 5534 | 89.01 | \ | \ |
| PIRATE | 3090 | 32.01 | 5408 | 30.02 |
| PIRATE+ | 3314 | 49.47 | 8866 | 1821.90 |

Table 7: Acronyms and corresponding full names in this paper.

| Acronyms | Full name | Acronyms | Full name |
|---|---|---|---|
| DIR | deformable image registration | DL | deep learning |
| CNN | convolutional neural network | PIRATE | plug-and-play image registration network |
| DEQ | deep equilibrium models | MBDL | model-based deep learning |
| PnP | plug-and-play | NCC | normalized cross-correlation |
| GCC | global cross-correlation | DSC | Dice similarity coefficient |
| JD | Jacobian determinant | MSE | mean squared error |
| STN | spatial transform network | FCN | fully convolutional network |
| DU | deep unfolding | SD-RED | steepest descent regularization by denoising |
| AWGN | additive white Gaussian noise | NODE | neural ordinary differential equations |
| ASM | adjoint sensitivity method | MMSE | minimum mean squared error |
| JFB | Jacobian-Free deep equilibrium models | ROI | region of interest |
| TV | total variation | | |

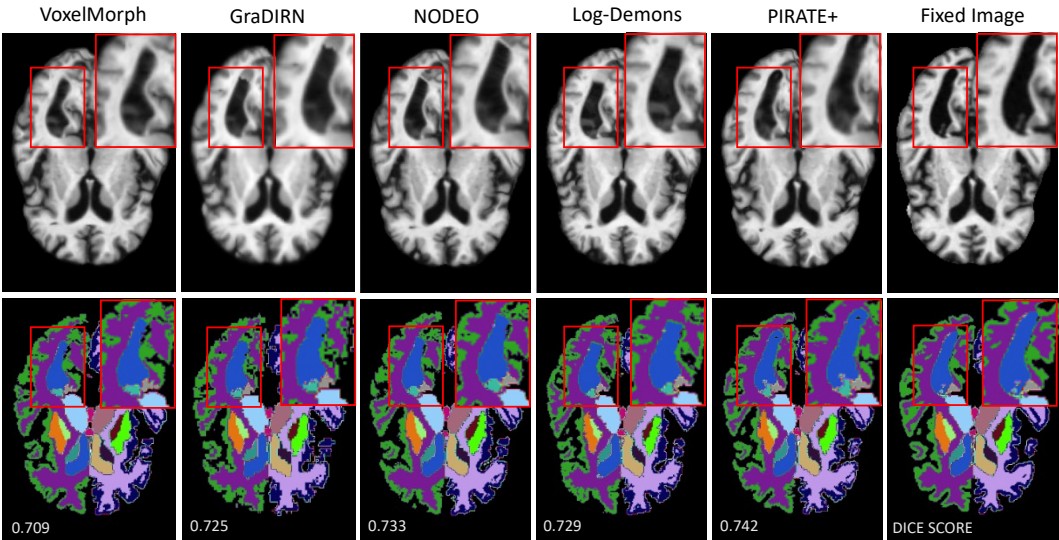

Figure 12: The visual results of warped images (top) and correlated warped segmentation masks (bottom) from the corner case of PIRATE+ (with the least improvement (*not* absolute value) in Dice) compared with results from other benchmarks on this case. The regions of interest are highlighted within a red box. The DSC for each image is displayed in the bottom row. Note that the result of PIRATE+ is still more consistent with the fixed image on this corner case.

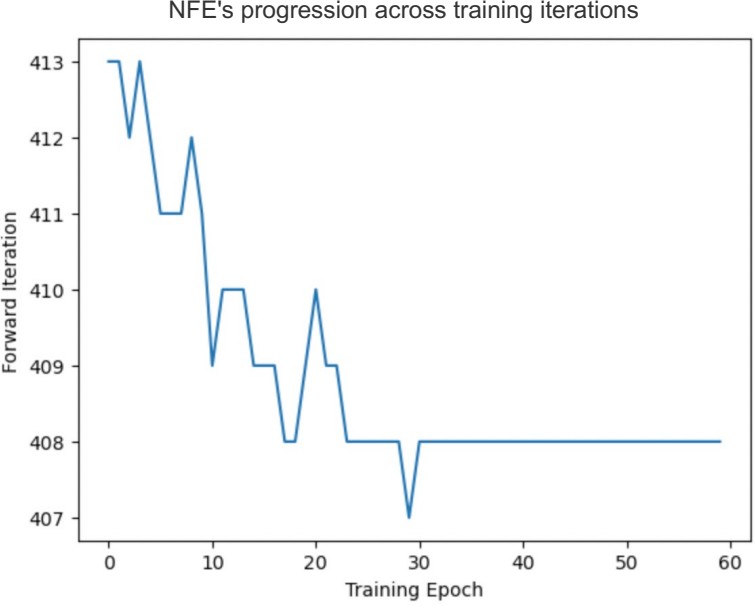

Figure 13: The plot for NFE along with the training iterations of PIRATE+. The results show a convergence behaviour.

