# OpenReview forum: "A Plug-and-Play Image Registration Network"
_ICLR.cc/2024/Conference — ICLR 2024 poster_

### Official Review · Reviewer_UpRR · 2023-10-29

**Soundness:** 4 excellent
**Presentation:** 3 good
**Contribution:** 4 excellent
**Rating:** 8
**Confidence:** 4

**Summary:**

The paper introduces a new deformable image registration (DIR) method called Plug-and-play Image Registration Network (PIRATE). PIRATE offers a new approach to DIR by integrating explicit data fidelity and a CNN prior. The paper also presents an extended version, PIRATE+, that fine-tunes the CNN prior using deep equilibrium models (DEQ). Both methods are validated on the OASIS and CANDI datasets, with results indicating state-of-the-art performance in DIR.

**Strengths:**

- Deformable image registration is a challenging task in medical image analysis, and the authors' approach of using a plug-and-play method to address this challenge is commendable.

- There is a comprehensive coverage of related work, which provides a solid foundation and context for the proposed method.

- The introduction of learned denoisers for regularizing the registration fields and the use of DEQ to fine-tune the regularizer within PnP iterations are innovative contributions.

- The extensive validation on two widely used datasets, OASIS and CANDI, highlights the robustness and general applicability of the proposed methods.

- The qualitative visual results presented in the paper convincingly demonstrate the superiority of PIRATE and PIRATE+ compared to existing deep learning and iterative methods.

**Weaknesses:**

- The incremental improvements in results, especially in the second decimal place, raise concerns about the practical implications of such minor improvements, particularly for downstream tasks, especially that the proposed method is an iterative optimization-based approach that significantly increases inference time compared to deep learning based methods.

- The paper makes heavy use of acronyms, which affects readability. Notably, the acronym "DU" is mentioned without a clear definition.

- While the paper's focus on brain MRI datasets is appreciated, it would have been beneficial to see the adaptability of PIRATE and PIRATE+ to other anatomies and imaging modalities.

**Questions:**

- How do the minor improvements in quantitative results translate to real-world applications, especially considering the potentially longer inference time of the iterative optimization-based approach?

- Is registration performed on the full 3D scan, or are 2D slices of roughly pre-aligned images used?

- How does PIRATE and PIRATE+ compare in terms of computational efficiency and scalability?

---

> ### Author Response · Authors · 2023-11-21
> **Response to reviewer UpRR**
>
> Thanks for your feedback on our work. Please see below for our point-by-point response to your comments.
>
> >1. (Weakness) The incremental improvements in results, especially in the second decimal place, raise concerns about the practical implications of such minor improvements, particularly for downstream tasks, especially that the proposed method is an iterative optimization-based approach that significantly increases inference time compared to deep learning based methods.  / (Q1) How do the minor improvements in quantitative results translate to real-world applications, especially considering the potentially longer inference time of the iterative optimization-based approach?
>
> We would like to highlight that PIRATE+ can gain noticeable improvement (more than 3% improvement in Dice score) than DL base methods (VoxelMorph, SYMNet, and GraDIRN). Compared to another iterative method, NODEO, PIRATE+ achieved slightly better results with  50% less inference time. Prompted by your comments, we also presented the best and the worst cases of registration results in Figure 2 and Figure 12, respectively. Note how PIRATE+ can gain visual improvement than the baseline, especially for small anatomical structures in the segmentation mask, in both the cases.
>
> >2. (Weakness) The paper makes heavy use of acronyms, which affects readability. Notably, the acronym "DU" is mentioned without a clear definition.
>
> Prompted by your comment, the revised manuscript listed all acronyms and corresponding full names in Table 7.
>
> >3. (Weakness) While the paper's focus on brain MRI datasets is appreciated, it would have been beneficial to see the adaptability of PIRATE and PIRATE+ to other anatomies and imaging modalities.
>
> Prompted by your comment, we conducted additional experiments on the suggested EMPIRE10 lung CT dataset. The revised manuscript reports the numerical result in Table 4 and visual results in Figure 6 and 7. These results showed that PIRATE and PIRATE+ also had better performance than other baselines on the new lung CT dataset.
>
> >4. (Q2) Is registration performed on the full 3D scan, or are 2D slices of roughly pre-aligned images used?
>
> We performed 3D registration: Both PIRATE and PIRATE+ performed on full 3D scans. All figures in the manuscript show a 2D slice of the registered 3D volumes.
>
> >5. (Q3) How does PIRATE and PIRATE+ compare in terms of computational efficiency and scalability?
>
> Prompted by your comment, we tested the computational efficiency and scalability of PIRATE and PIRATE+ in Table 6 and Figure 11. Table 6 shows that PIRATE+ has higher computational and memory complexity than PIRATE. Figure 11 shows that PIRATE+ and PIRATE has similar scalability in terms of time and memory cost for inference.

---

> ### Author Response · Authors · 2023-11-22
> **Response to Reviewer UpRR**
>
> Dear Reviewer UpRR, thank you again for your positive comments on our paper. Please let know if there is anything else we can address that would help to improve your evaluation of our paper.

---

> > ### Comment · Reviewer_UpRR · 2023-11-22
> >
> > Thanks to the authors for their detailed and insightful response to the concerns I raised. After thoroughly considering your explanations and reflecting on the input from other reviews, I have decided to increase my original score.

---

> > > ### Author Response · Authors · 2023-11-22
> > > **Response to Reviewer UpRR**
> > >
> > > Thank you for raising your score and your positive comments.

---

### Official Review · Reviewer_3727 · 2023-10-30

**Soundness:** 3 good
**Presentation:** 3 good
**Contribution:** 3 good
**Rating:** 8
**Confidence:** 4

**Summary:**

The paper addresses the problem of deformable (non-rigid) image registration in the context of biomedical image analysis. In particular, it proposes two methods (PIRATE and PIRATE+) for the regularisation of the deformation field. In contrast to existing deep learning approaches to image registration, the approach presented here explicitly integrates a data-fidelity penalty as well as a CNN prior (this is pre-trained and acts as regularizer for the deformation field). The authors argue that this improves the fidelity between the registered image and the reference image. The second approach, PIRATE+, is similar to this but uses a CNN prior that is trained using deep equilibrium models.

**Strengths:**

The proposed methods (PIRATE and PIRATE+) are novel methodological contributions which is positive. Additionally, the proposed framework is compared to a number of different methods, including DL and non-DL methods, which is very good. This also includes the best-performing method from recent comparative studies (Mok and Chung, CVPR 2020, MICCAI 2020). The results reported outperform this method in terms of registration accuracy. Another strength of the paper is the careful review of the state-of-the-art in the field. This is well done and comprehensive, allowing the reader to place the proposed work in the context of the SOTA.

**Weaknesses:**

The weaknesses are mainly related to the evaluation of the proposed framework:

- The methods compared in the paper use very different loss or cost functions as well as different models for the parameterization of the deformation field. This makes the papers' comparison of the registration accuracy in terms of voxels with negative Jacobian very difficult to come across methods. Registration accuracy measured in terms of Dice is more meaningful. At the same time, it would have been good if the authors had used some additional non-brain datasets which have landmarks and thus allow the calculation of quantities such as the target registration error. One such dataset is from the EMPIRE10 challenge...

- The run-time of the proposed framework is significantly higher than those of other DL methods. This is a significant disadvantage for clinical applications.

**Questions:**

- The paper proposes two methods, PIRATE and PIRATE+. I am a bit unclear on what are the conclusions: Is PIRATE+ is better than PIRATE? When should PIRATE be used? When should PIRATE+ be used?

- How are the parameters in the other registration methods chosen, especially in the context of trading off registration accuracy and regularisation of the deformation field?

- The run-time of the proposed framework is significantly higher than those of other DL methods. What are the reasons for this?

**Details Of Ethics Concerns:**

No ethics concerns

---

> ### Author Response · Authors · 2023-11-21
> **Response to reviewer 3727**
>
> Thanks for your feedback on our work. Please see below for our point-by-point response to your comments.
>
> >1. (Weakness) The methods compared in the paper use very different loss or cost functions as well as different models for the parameterization of the deformation field. This makes the papers' comparison of the registration accuracy in terms of voxels with negative Jacobian very difficult to come across methods. Registration accuracy measured in terms of Dice is more meaningful. At the same time, it would have been good if the authors had used some additional non-brain datasets which have landmarks and thus allow the calculation of quantities such as the target registration error. One such dataset is from the EMPIRE10 challenge...
>
> Prompted by your comment, we conducted additional experiments on the suggested EMPIRE10 lung CT dataset. The revised manuscript reports the numerical result in Table 4 and visual results in Figure 6 and 7. These results show that PIRATE and PIRATE+ can maintain better performance than other baselines on the new lung CT dataset.
>
> >2. (Weakness) The run-time of the proposed framework is significantly higher than those of other DL methods. This is a significant disadvantage for clinical applications.  / (Q3) The run-time of the proposed framework is significantly higher than those of other DL methods. What are the reasons for this?
>
> The inference time of PIRATE and PIRATE+ are longer than other DL based methods (VoxelMorph, SYMNet, and GraDIRN) due to their iterative scheme. However, our results show that PIRATE+ can gain noticeable improvement (more than 3% improvement in Dice score) than those DL methods. Moreover, the testing time of PIRATE and PRIATE+ (around 40 seconds per 3D volume) is still negligible in clinical applications when real-time registration is not needed. One such application is to generate aligned image pairs required for training image translation models. Note also that our approach does not focus on real-time clinical application at the current stage.
>
> >3. (Q1) The paper proposes two methods, PIRATE and PIRATE+. I am a bit unclear on what are the conclusions: Is PIRATE+ is better than PIRATE? When should PIRATE be used? When should PIRATE+ be used?
>
> Our results show that PIRATE+ outperforms PIRATE with a designated iterative algorithm when trained to optimize the task-specific algorithmic loss. The choice between PIRATE and PIRATE+ depends on the availability of training datasets and trade-off between performance and training complexity. In PIRATE+, one needs pairs of MRI data for training and to optimize a sophisticated DEQ loss related to a specific registration task. On the other hand, PIRATE only requires a set of registration fields, and the training loss corresponds to a simple regression task. The revised manuscript also reported the training complexity of PIRATE and PIRATE+ in Table 6.
>
> >4. (Q2) How are the parameters in the other registration methods chosen, especially in the context of trading off registration accuracy and regularisation of the deformation field?
>
> We used the official codes of other registration baselines with the suggested parameters (parameters on brain datasets). We retrained all models on our datasets to make fair comparisons.

---

### Official Review · Reviewer_yfCW · 2023-10-31

**Soundness:** 2 fair
**Presentation:** 2 fair
**Contribution:** 2 fair
**Rating:** 6
**Confidence:** 1

**Summary:**

They proposed a new DIR method which is the plug-and-play approach that trains a CNN-based denoiser on the registration field. With
this denoiser as a regularizer within iterative methods, deep equilibrium learning is used as a fixed point iterator. The authors use a pre-trained denoiser for such regularization problems. The proposed method achieved the best performances on OASIS & CANDI datasets with reasonable qualitative results corresponding to quantitative results.

**Strengths:**

+ The proposed method used the DEQ approach for iterative registrations. The adapting approach is unique and reasonable. Their training loss looks going down well on the datasets.
+ DEQ approach successfully addressed registration problems with PnP(plug-and-play) method with appealing gain in Tables 1 & 2.

**Weaknesses:**

+ I didn't understand the motivation of using pre-trained denoisor for regularizations. Why isn't the trainable denoiser used for this task?
+ No ablation studies about the necessity of the usage of a pre-trained model
+ Less analysis on the DEQ model : I would like to see the convergence of the number of function evaluation(NFE) along with the training iterations.
+ In general, the result section is short of significant experiments to support their claim. For example, Figure 3 is a single instance analysis rather group analysis.
+ Generalization needs to be validated with group analysis.

**Questions:**

+ What is the corner case for the proposed method?
+ Memory consumption needs to be reported according to fixed size of datasets.

---

> ### Author Response · Authors · 2023-11-21
> **Response to reviewer yfCW**
>
> Thanks for your feedback on our work. Please see below for our point-by-point response to your comments.
>
> >1. (Weakness) I didn't understand the motivation of using pre-trained denoisor for regularizations. Why isn't the trainable denoiser used for this task? No ablation studies about the necessity of the usage of a pre-trained model.
>
> Motivation of using pre-trained denoiser: The recent literature has shown the potential of training neural network denoisers to model the statistical distribution of high-dimensional data (e.g., denoising diffusion models and plug-and-play priors). Our work is the first to adopt this idea for registration by training a **denoiser prior** over the registration fields. The revised manuscript has clarified the choice of using denoiser prior for image registration.
>
> Why not trainable denoiser: We would like to highlight that we did use trainable denoiser in PIRATE+. In PIRATE+, we train the denoiser on a dataset by using the loss specific to the registration task computed using the registration field, the warped image and the fixed image.
>
> >2. (Weakness) Less analysis on the DEQ model : I would like to see the convergence of the number of function evaluation(NFE) along with the training iterations.
>
> Prompted by your comment, we tested NFE across training iterations in DEQ. The results are shown in **Figure 13 in the supplement**. Figure 13 shows that the NFE converges during the training process.
>
> >3. (Weakness) In general, the result section is short of significant experiments to support their claim. For example, Figure 3 is a single instance analysis rather than group analysis. Generalization needs to be validated with group analysis.
>
> We would like to highlight that **we did conduct group analysis in Table 1, 2, and 3 in our original manuscript**. In those tables, we reported the mean and standard deviation values of the entire testing dataset, which consists of 100 pairs of 3D MRI volumes.
>
> >4. (Q1) What is the corner case for the proposed method?
>
> Prompted by your comment, the revised manuscript shows the best (with the most dice improvement, **not** absolute Dice value) and the worst case (with the least dice improvement) in Figure 2 and Figure 12, respectively. In both cases, PIRATE+ can gain better performance than other baseline models.
>
> >5. (Q2) Memory consumption needs to be reported according to fixed size of datasets.
>
> Prompted by your comment, the revised manuscript shows the memory consumption on both training and inference stages of PIRATE, PIRATE+ and other baselines in Table 6. Note that PIRATE and PIRATE+ has lower memory complexity compared to other DL baselines.

---

> > ### Comment · Reviewer_yfCW · 2023-11-22
> >
> > Thanks for the efforts to address previous queries. I believe major parts of mentioned queries are properly addressed, so I raised my score.

---

> > > ### Author Response · Authors · 2023-11-22
> > > **Response to Reviewer yfCW**
> > >
> > > Thank you for reading our response and raising the score.

---

### Official Review · Reviewer_kX3w · 2023-11-02

**Soundness:** 3 good
**Presentation:** 3 good
**Contribution:** 3 good
**Rating:** 6
**Confidence:** 3

**Summary:**

This paper introduces a new deformable registration framework for medical imaging. Their main contribution is the inclusion of a "plug and play" prior into the registration framework. A novelty of the work is using a denoiser to specify priors over registration fields. They also propose an additional model (PIRATE+) that fine-tunes the CNN prior in PIRATE using deep equilibrium models (DEQ). The authors then evaluate their work on standard brain MRI datasets used in the registration literature.

**Strengths:**

The paper seems to be the first at using plug-and-play priors for registration, which could be a useful contribution in this already rich space.

The work obtains good quantitative results on standard benchmarks for deformable medical image registration.

**Weaknesses:**

My main issue is with the writing and presentation of this paper:

- There was virtually no intuitive explanation of what plug-and-play does, why it is useful compared to other approaches, etc. Given that one of the reported contributions of the paper is to show that denoising priors can be used for registration, the lack of any explanation of why denoising is appropriate, how it works, etc. is quite conspicuous.

- What is the main drawback with current deformable image registration models that this current approach is addressing? Why might plug and play be better than other forms of priors for registration? The introduction does not address this.

- The methods section is poorly written, without high-level insights presented before details. For example, the methods starts immediately with the PIRATE iteration updates instead of presenting what the objective being optimized is, what the prior is capturing, etc.

- It is unclear what issue PIRATE+ is trying to improve upon. The methods section simply says "PIRATE+ uses DEQ to fine-tune the regularizer D in the PIRATE iteration by minimizing...", but there was no explanation of why the regularizer might not be accurate in the first place.

- The method section would benefit by using pointers back to Fig. 1 and explaining the various steps of that figure.

**Questions:**

1. What advantages do you think plug and play priors offer for registration ?

2. Why might denoising be appropriate as a prior for registration?

3. I am not able to understand what PIRATE+ addressing? The methods says "PIRATE+ fine-tunes the AWGN denoiser into a task-specific regularizer using DEQ" but this is quite opaque to me. What does "task-specific regularization" mean in this case?

4. Are there any limitations of the method compared to traditional CNN-based deformation models like VoxelMorph in terms of usability, training time, etc.?

---

> ### Author Response · Authors · 2023-11-21
> **Response to reviewer kX3w**
>
> Thanks for your feedback on our work. Please see below for our point-by-point response to your comments.
>
> >1. (Weakness) There was virtually no intuitive explanation of what plug-and-play does, why it is useful compared to other approaches, etc. Given that one of the reported contributions of the paper is to show that denoising priors can be used for registration, the lack of any explanation of why denoising is appropriate, how it works, etc. is quite conspicuous. Why might denoising be appropriate as a prior for registration?  / (Q1) What advantages do you think plug and play priors offer for registration ? / (Q2) Why might denoising be appropriate as a prior for registration?
>
> The recent literature has shown the potential of training neural network denoisers to model the statistical distribution of high-dimensional data (e.g., denoising diffusion models and plug-and-play priors). Our work is the first to adopt this idea for registration by training a **denoiser prior** over the registration fields. The revised manuscript has clarified the choice of using denoiser prior for image registration. Please also refer to our analysis in section 3.1.
>
> >2. (Weakness) What is the main drawback with current deformable image registration models that this current approach is addressing? Why might plug and play be better than other forms of priors for registration? The introduction does not address this. / (Q1) What advantages do you think plug and play priors offer for registration?
>
> PIRATE and PIRATE+ do not focus on addressing issues in existing DL methods. Our approach instead is a novel optimization method for image registration that leverages a CNN-based prior over registration fields. Our paper shows superior performance using the learned prior compared to handcrafted priors (such as smoothness), while also achieving the SOTA performance compared to the DL methods.
>
> >3. (Weakness) The methods section is poorly written, without high-level insights presented before details. For example, the methods starts immediately with the PIRATE iteration updates instead of presenting what the objective being optimized is, what the prior is capturing, etc.
>
> Prompted by your comment, we added high-level insights of PnP/Denoiser prior in the introduction. We also included the objective function in the method section of the revised manuscript as below.
>
> The object function that PIRATE optimizes is formulated as a variation of (1)
>
> $\pmb{\hat{\phi}} = \arg \mathop{\min}\limits_{\pmb{\phi}} \lbrace g(\pmb{f},\pmb{\phi} \circ \pmb{m}) + h(\pmb{\phi}) \rbrace$
>
> where $h$ represents the regularizer consisting of an explicit smoothness constraint and the implicit denoising regularizer. The iteration of PIRATE can be expressed as……..
>
> >4. (Weakness) The method section would benefit by using pointers back to Fig. 1 and explaining the various steps of that figure.
>
> We added a pointer back to Fig. 1 in section 3 paragraph 1 of the revised manuscript.
>
> >5. (Weakness) It is unclear what issue PIRATE+ is trying to improve upon. The methods section simply says "PIRATE+ uses DEQ to fine-tune the regularizer D in the PIRATE iteration by minimizing...", but there was no explanation of why the regularizer might not be accurate in the first place. / (Q3) I am not able to understand what PIRATE+ addressing? The methods says "PIRATE+ fine-tunes the AWGN denoiser into a task-specific regularizer using DEQ" but this is quite opaque to me. What does "task-specific regularization" mean in this case?
>
> The regularizer in PIRATE+ is learned by minimizing **the loss specific to the registration task computed using the registration field, the warped image and the fixed image**. On the other hand, the regularizer in PIRATE is learned by minimizing MMSE loss between denoised registration fields and noisy registration fields, without incorporating any specific information about registration tasks. For the specific task PIRATE+ is trained on, PIRATE+ can gain improved performance compared to PIRATE. The revised manuscript has highlighted this difference.
>
> >6. (Q4) Are there any limitations of the method compared to traditional CNN-based deformation models like VoxelMorph in terms of usability, training time, etc.?
>
> The training time of PIRATE+ is longer than other traditional CNN-based methods. The additional training time comes from the fixed point iterations of PIRATE. However, our results show that PIRATE+ can gain noticeable improvement (more than 3% improvement in Dice score) than traditional CNN-based methods. The revised manuscript has clarified this limitation in the conclusion section.

---

> ### Author Response · Authors · 2023-11-22
> **Ending of the discussion period**
>
> Dear Reviewer kX3w, as we are nearing the end of the discussion period, please let us know if there is anything else we can address to improve your evaluation of our paper.

---

> > ### Comment · Reviewer_kX3w · 2023-11-23
> > **Revised Score**
> >
> > Thanks for the clarifications. I have increased my score accordingly.

---

> > > ### Author Response · Authors · 2023-11-23
> > > **Response to reviewer kX3w**
> > >
> > > Thank you for raising your score and your positive comments.

---

### Author Response · Authors · 2023-11-21
**Response to all reviewers**

Thank you all for your valuable feedback. We provide point-to-point responses to all the comments below. In summary, we clarified our contribution of using denoisers as priors for image registration, ran additional experiments on a new lung CT dataset, and tested the computational efficiency and scalability of our method. We highlighted changes in red in the revised manuscript and the supplement.

---

### Author Response · Authors · 2023-11-22
**Comment to all reviewers and area chairs**

Dear all, we are nearing the end of the discussion period. We have responded to all the comments. We hope that our responses will help other reviewers to also see the value in our work.

We are enthusiastic about this work due to multiple reasons: (a) we propose PIRATE as the **first plug-and-play methods for image registration problem**; In PIRATE, we train a deep **denoiser to model the statistical distribution of the registration fileds** and use it as a prior under optimization framework, (2) we propose **PIRATE+ to further improve the performance of PIRATE** by using deep equilibrium models to fine-tune the denoiser, and (3) our numerical validations demonstrate **SOTA performance**.

Please let us know if there is anything else we can clarify or answer until the deadline.

---

### Meta-Review · Area_Chair_Y2rE · 2023-12-06

**Metareview:**

This paper presents a plug and play image registration network that trains a CNN-based denoiser on the registration field. Overall, the paper is well written and the method is sound. Although the motivation of using  a CNN-based denoise as regulizer is not clearly explained, the fidelity is indeed an issue in deformable image registration of brain MRI. The authors' main contribution is the introduction of  training neural network denoisers to the specific image registration task, which has reasonable contribution. The rebuttal has been effective and addressed the main concerns from the reviewers.

Although I am overall supportive, I felt that the title of the paper shall be modified to reflect the fact that it is only validated on brain MRI or at least medical image, say "A Plug-and-Play Brain MRI Registration Network".

**Justification For Why Not Higher Score:**

The paper has some contribution, but the idea is not that 'novel' for higher scores.

**Justification For Why Not Lower Score:**

The authors mainly introduce CNN denoiser to deformable image registration, which in my opinion has some contribution. But it could be possible that some readers may consider this as "obvious" or limited novelty.

---

### Decision · Program_Chairs · 2024-01-16

Accept (poster)